# Agomir-331 Suppresses Reactive Gliosis and Neuroinflammation after Traumatic Brain Injury

**DOI:** 10.3390/cells12202429

**Published:** 2023-10-11

**Authors:** Jin-Xing Wang, Xiao Xiao, Xuan-Cheng He, Bao-Dong He, Chang-Mei Liu, Zhao-Qian Teng

**Affiliations:** 1State Key Laboratory of Stem Cell and Reproductive Biology, Institute of Zoology, Chinese Academy of Sciences, Beijing 100101, Chinahexuancheng@ioz.ac.cn (X.-C.H.);; 2Savaid Medical School, University of Chinese Academy of Sciences, Beijing 100408, China; 3Institute for Stem Cell and Regeneration, Chinese Academy of Sciences, Beijing 100101, China; 4Beijing Institute for Stem Cell and Regenerative Medicine, Beijing 100101, China

**Keywords:** traumatic brain injury, miR-331, IL-1β, agomir, glial scar, neuroprotection

## Abstract

Traumatic brain injury usually triggers glial scar formation, neuroinflammation, and neurodegeneration. However, the molecular mechanisms underlying these pathological features are largely unknown. Using a mouse model of hippocampal stab injury (HSI), we observed that miR-331, a brain-enriched microRNA, was significantly downregulated in the early stage (0–7 days) of HSI. Intranasal administration of agomir-331, an upgraded product of miR-331 mimics, suppressed reactive gliosis and neuronal apoptosis and improved cognitive function in HSI mice. Finally, we identified IL-1β as a direct downstream target of miR-331, and agomir-331 treatment significantly reduced IL-1β levels in the hippocampus after acute injury. Our findings highlight, for the first time, agomir-331 as a pivotal neuroprotective agent for early rehabilitation of HSI.

## 1. Introduction

Traumatic brain injury (TBI) triggers glial scar formation and progressive neuroinflammation, two intertwined key pathological processes underpinning neurodegeneration [1]. The glial scarring is characterized by reactive astrogliosis and microglia around the injury core, which largely inhibit axonal growth and remyelination [2,3]. Reactive astrocytes are pivotal responders to all forms of TBI, which exhibit hypertrophy and increased immunoreactivity of glial fibrillary acidic protein (GFAP) [4]. Astrocytes within the scars may prevent the neural recovery after injury by secreting growth-inhibitory molecules, such as semaphorin 3A [5]. Microglia play an essential role in the brain’s immune system by dynamically surveying the brain environment. Microglia react to TBI within minutes and undergo morphological and functional changes to elicit pro- or anti-inflammatory responses [6]. Progressive neuroinflammation is propagated mainly by microglia. Pro-inflammatory microglia predominate at the injury site and release cytokines such as IL-1β, IL-6, and TNF-α; on the other hand, anti-inflammatory microglia produce IL-10, TGF-β, NGF, BDNF, and GDNF, which promote neuroprotection and regeneration [7,8]. Microglia are activated earlier than astrocytes and induce astrocytic activation, while activated astrocytes accelerate microglial activation and inhibit microglial activities under various pathological conditions [9]. Given that glial scarring and neuroinflammation are often correlated with neuropathology, modulating these pathological processes will aid in developing therapies to combat death and disability after TBI [3,4].

MicroRNAs (miRNAs) are a class of small non-coding RNAs that function in posttranscriptional gene regulation by binding to the 3′-untranslated regions of target genes [10]. Analysis of the brain miRNAome demonstrates that many miRNAs are up- or down-regulated after TBI [11,12]. Moreover, several studies have reported that miRNA mimics or inhibitors are able to improve outcomes in rodent models of neurological diseases [13,14,15], indicating that targeting miRNAs might be a promising therapeutic strategy for the treatment of disease.

MiR-331 is expressed abundantly in the brain, especially in neurons, astrocytes, and microglia [16,17]. Previous studies have reported that miR-331 suppresses cell migration in both colorectal carcinoma [18] and rheumatoid arthritis [19,20], and alleviates lipopolysaccharide-induced damage of pc12 cells [19,20]. Importantly, miR-331 is one of the most down-regulated miRNAs in the brain within 12 h of TBI [21], but its roles in the pathogenesis of TBI remain to be elucidated.

This study provides evidence that intranasal administration of miR-331 mimics represses reactive gliosis and neuroinflammation, and improves cognitive function in a mouse model of hippocampal stab injury (HSI). Mechanistically, interleukin-1 beta (IL-1β) is a direct downstream target of miR-331 following HSI.

## 2. Materials and Methods

### 2.1. Animal

The Animal Committee of the Institute of Zoology Chinese Academy of Sciences approved all the animal procedures in our studies. Male adult mice (6–8 weeks old; body weight 19–22 g), with C57BL/6J genetic background, were purchased from a commercial supplier, Beijing SPF Biotechnology Co., Ltd. (Beijing, China) Mice were raised in an environmentally controlled chamber with a temperature of 22 ± 3 °C and a 12 h dark/12 h light cycle.

### 2.2. Blade Penetrating Stab Wound

The mouse model of TBI was generated using a blade-penetrating stab wound to the hippocampus as described previously [22,23]. In brief, male mice were intraperitoneally (i.p.) anesthetized with Avertin (200 mg/kg, T48402, Sigma-Aldrich, St. Louis, MO, USA), and the mouse’s head was positioned in a KOPF stereotaxic apparatus. To expose the skull, a skin incision, 1.0 cm in length, was performed along the midline of the skull. On the right side of the brain, a skull cranial window (1 mm wide and 4 mm long) was made using a #12 scalpel blade. Then a sterile scalpel blade #15 was inserted 3.5 mm deep in the brain to make a hippocampal lesion. The coordinates (from bregma) of the blade penetrating stab are listed as follows: AP (posterior-posterior) = 1.6 mm, ML (mediallateral) = 0.9 mm, AP = 4.5 mm, ML = 2.6 mm. After insertion, the blade remained in the tissue for 1 min to avoid blood backflow and was slowly removed. After completing all injury procedures, the skin incision was closed with absorbable sutures. Once the mice were fully recovered, they were returned to their home cages.

### 2.3. Intranasal Delivery of Agomir-331

Immediately after TBI, agomir-331 or scramble nucleotides were intranasally delivered to the brain every other day for 3 or 7 days, as described previously [23,24]. At each time point of drug administration, a dose of 1 nmol of agomir-331 or scramble (GenePharma Biotech, Shanghai, China) was dissolved in 24 μL RNase-free water and intranasally delivered in 6 drops (4 µL/drop), alternating between each nostril.

### 2.4. Cell Transfection

Cell transfections were conducted using Lipofectamine 3000 reagent (Invitrogen, Carlsbad, CA, USA) following the manufacturer’s instructions. The agomir-331 or scramble nucleotides (20 μM) and transfection reagent were mixed with the Opti-MEM medium and incubated for 15 min. The final solution was then transfected into microglia or astrocytes.

### 2.5. Primary Astrocyte and Microglia Culture

As previously described, isolation and purification of primary microglia or astrocytes from c57bl/6 mice were performed at postnatal day 1–2 [25]. We cut the cortical tissue into small pieces (approximately 1 mm^3^), digested them with TrypLE and DNase I for 8–10 min, and stopped the digestion with complete DMEM medium (10% FBS and 1% penicillin/streptomycin). We filtered the cell suspension through a 70 µm cell strainer and seeded cells in a culture flask. After 12–14 days of culture, the flasks containing mixed glia (astrocyte-microglia) were shaken at 200 rpm for 50 min to collect primary microglia suspension, and the cells were re-cultured in 12- or 24-well plates. Oligodendrocytes were removed by continuously shaking at 200 rpm for 3 h, and purified astrocytes (purity > 90%), adherent cells in the flask, were cultured for further analysis.

Microglial conditioned medium (MCM) was collected to simulate the microenvironment of TBI in vitro as previously described [26,27]. Briefly, mouse primary microglia were seeded in 10 cm dishes, and the cells were stimulated with lipopolysaccharide (LPS, 100 ng/mL) for 6 h and washed with phosphate buffered saline (PBS) to avoid LPS contamination. Next, primary microglia were cultured with a complete medium for another 24 h; after that, MCM was collected and filtered through a 0.22 μm diameter filter.

### 2.6. Quantitative Real-Time PCR (qRT-PCR)

To dissect the hippocampal or cortex tissues, mice were transcardially perfused with cold PBS for 5 min after deep anesthesia. The hippocampus and cortex were carefully separated in cold PBS using capsulorhexis forceps. Cortical or hippocampal tissue samples approximately 2 mm wide were collected along the wound track under a microscope and rapidly frozen in liquid nitrogen and stored at −80 °C until use for RNA extraction.

Following the manufacturer’s instructions, total RNA was extracted from microglia or hippocampal tissue using TRIzol (Invitrogen, Carlsbad, CA, USA). RNA quality and concentration were determined with a NanoDrop 2000 spectrophotometer. A quantity of 1 μg RNA was then reverse-transcribed into cDNA using the Transcription First Strand cDNA Synthesis Kit (TransGen Biotech, Beijing, China). To examine the expression levels of genes, qRT-PCR was performed using the SYBR^®^ Green assay (Yeasen, Shanghai, China) [28]. U6 and β-actin served as endogenous control genes for miRNA and mRNA expression, respectively. The relative gene expression was measured via the 2^−ΔΔCt^ method. qRT-PCR primers are listed in Table 1.

### 2.7. Immunostaining

After deep anesthesia, mice were transcardially perfused with cold PBS for 5 min, followed with cold 4% paraformaldehyde (PFA) for 5 min. Harvested mouse brains were postfixed in 4% PFA (at least 12 h, 4 °C), then equilibrated in 30% sucrose for dehydration (48 h, 4 °C). Next, brains were then sectioned into 40 μm thick serial sections using a cryostat. Immunohistochemical staining was performed on one in three floating sections containing hippocampus and cortex from 4 mice per group, as previously described [29]. Sections were washed with 1× PBS three times (10 min each), and then incubated in a blocking solution containing 2% BSA and 0.3% Triton X-100 for 2 h at room temperature. Then, the sections were incubated with primary antibodies overnight at 4 °C in a humidified chamber. After three washes with 1× PBS (pH 7.4), sections were incubated with secondary antibodies conjugated to Alexa Fluor 488, 568, or 647, and DAPI at room temperature for 1–2 h. Lastly, the PBS-washed brain sections were mounted on glass slides, embedded with adhesion anti-fade medium, and stored at −20 °C in the dark. Four 20×x micrographs were collected for each section, and two squares of 200 × 200 µm around the lesion core were analyzed for each micrograph.

In BrdU incorporation assay, primary astrocytes were transfected with agomir-331 or scramble and exposed to MCM for 24 h. A quantity of 5 µM BrdU was then added to the culture medium for 6 h, and the astrocytes were fixed in 4% PFA for 20 min and subsequently washed 3 times with 1× PBS. The cells were soaked in 1 M HCL for 20 min at 37 °C, washed with 0.1 M sodium tetraborate buffer (pH 8.5), and finally immunostained with anti-BrdU.

The primary antibodies were as follows: anti-BrdU (ab6326, Abcam, Waltham, MA, USA; 1:500), anti-GFAP (16825-1-AP, Proteintech, Rosemont, IL, USA; 1:1000), anti-IBA1 (NB100-1028, NOVUS, St. Charles, MO, USA; 1:1000), and anti-NeuN (MAB377, Millipore, Burlington, MA, USA; 1:1000). Alexa Fluor conjugated secondary antibodies were used at a dilution of 1:500.

### 2.8. Cell Viability Assay

Cell Counting Kit-8 (CCK-8, Beyotime, Nantong, China) was used to detect cell viability. The cells were seeded in 96-well plates at a density of 10^5^/mL. A quantity of 10 μL of CCK-8 solution was added to each well of the plate. After 2 h of incubation in a cell culture incubator, the absorbance value of the samples was determined at 450 nm.

### 2.9. Wound Healing Scratch Assay

Wound healing scratch assay was performed as described previously [30]. The cells were seeded into 12-well plates. After 24 h, the cells were scratched with a 200 μL pipette tip, and the cell debris was washed out with fresh medium. The scratched areas were imaged at given time points. Scratch closure was determined by measuring the change in the wound width over time. This method calculates the ratio of the width of the gap at a given period of time to the width of the initial scratch. The width is the average distance between the two margins of the gap.

### 2.10. TUNEL Assays

TUNEL staining was performed with the TUNEL kit (Beyotime, China) following the manufacturer’s instructions. After the immunohistochemical staining of NeuN, brain sections were incubated with a TUNEL mix (5 µL TdT enzyme diluted in 45 µL staining solution) for 1 h at 37 °C.

### 2.11. Behavioral Tests

The mice were randomly assigned to three groups: sham group, scramble-treated TBI group, and agomiR-331-treated TBI group. All tests were conducted during the light phase. The changes in cognitive function after TBI were assessed using memory tests in the Barnes maze and Y maze as described in previous studies [22,23].

The Barnes maze test was performed on a gray circular platform with a diameter of 120 cm and 20 equidistant holes (5 cm in diameter) at 9 to 13 days after injury. Paper pieces with different shapes were tipped to the walls as spatial cues for the mice in behavioral tests. The experimental process mainly consisted of three steps: habituation (Day 1), training (Days 2–3), and probe (Day 5). During the habituation process, the mice were gently placed in the platform’s center with a clear glass beaker and were given about 2 min to freely explore the maze. After 2 min, the tested mice were allowed to find the escaped cage and stay under guidance for about one minute. During the training phase, all mice were allowed to search for a hiding box starting from the maze’s center for 2 min. If the mice had not found the hiding box after 2 min, the mouse was manually guided to the hiding box. Three trials were performed by each mouse on the first day of training and two trials were performed on the second day of training. During the probe phase, mice freely explored the maze for 2 min after removing the hiding box.

The Y-maze consisted of three identical arms (50 × 10 × 15 cm), which diverged at a 120° angle. The Y-maze test was carried out at 15 to 16 days after injury. On the test day, the mouse was positioned in one arm and allowed to freely explore the maze for 10 min. The counts of entries into the 3 arms (total arm entries) and sequence were recorded. A spontaneous or successful alternation is defined as a mouse consecutively entering all three arms, and the percentage of spontaneous alternation is calculated as [spontaneous arm alternations/(total arm entries − 2)] × 100.

### 2.12. Enzyme-Linked Immunosorbent Assay (ELISA)

At 3 days after TBI and miR-331 agomir treatment, hippocampal tissues were homogenized with the protein lysis buffer, and the supernatants were extracted after centrifugation at 1000× *g*. A commercial ELISA kit was used to detect IL-1β expression following the manufacturer’s instructions, and the cytokine concentration was determined according to the standard curve.

### 2.13. Luciferase Reporter Assay

The luciferase reporter assay was carried out according to our previous protocol [23]. We used PCR to amplify the sequence (~400 bp) of IL-1β 3′ UTR incorporating the putative miR-331 binding site from genomic DNA, and cloned into pmirGLO (Promega, Madison, WI, USA; E1330). The primers for cloning of IL-1β 3′ UTR were as follows: IL-1β forward 5′-TGTTTCTAATGCCTCCCCAG-3′, IL-1β reverse: 5′-AAAGCAATGTGCTGTGCTC-3′. The miR-331 binding sites on the IL-1β 3′UTR was mutated using the Quick-Change II Site-directed Mutagenesis Kit (Stratagene, La Jolla, CA, USA) based on the manufacturer’s instructions. The primers for subcloning the mutated IL-1β 3′UTR were as follows: forward, 5′-AGACAGCTCAATCTGGGCCACTCCTTAGTCCTCG-3′; reverse, 5′-CGAGACTAAGGAGTGTGCCCAGATTGAGCTGTCT-3′. PmirGLO-3′UTR or mutated pmirGLO-3′UTR were co-transfected with miR-331 agomir or scramble control into HEK293 cells using Lipofectamine 3000 (Invitrogen, Carlsbad, CA, USA). We followed the manufacturer’s protocol to record all Luciferase readings using the Dual-Luciferase Reporter 1000 System (Promega, Madison, WI, USA).

### 2.14. Statistical Analysis

All the statistical analyses were performed using GraphPad Prism software (version 8.02, GraphPad Prism Software, San Diego, CA, USA). The Shapiro–Wilk test was applied to evaluate whether a dataset is normally distributed. In the present study, all datasets passed the normality test. The significance of differences was assessed by two-tailed, unpaired Student’s *t*-test or one-way ANOVA followed by Tukey’s multiple comparisons tests. *p* < 0.05 was considered statistically significant. All data are presented as means ± standard error of the mean (SEM).

## 3. Results

### 3.1. The Expression of miR-331 Is Downregulated in Brains with Early TBI

To determine whether miR-331 is dysregulated after TBI, we examined its expression in the cortex and hippocampus at different time windows after hippocampal stab injury (HSI) by qRT-PCR. miR-331 expression was significantly decreased in the cortex (*F*_(4,15)_ = 15.391, *p* < 0.001) at 1 to 7 days post injury (dpi) and recovered to sham level at 14 dpi (Figure 1A). Similarly, downregulation of miR-331 was also observed in the injured hippocampus (*F*_(4,15)_ = 18.468, *p* < 0.001) at 1 to 7 dpi (Figure 1B). These data suggested that miR-331 might play a role in the pathogenesis of disease and be a promising therapeutic target for the acute phase of HSI.

### 3.2. Supplementation with Agomir-331 Protects against Neuronal Apoptosis after HSI

To examine whether miR-331 is a therapeutic target for HIS, we synthesized agomir-331, an upgraded product of miR-331 mimics with a higher stability and inhibiting effect, and applied it to HSI mice at a dose of 1nM every other day for 3 or 7 days via the intranasal route. Our immunohistochemical staining results demonstrated that the administration of agomir-331 increased the number of NeuN^+^ neurons (*F*_(2,9)_ = 378.734, *p* < 0.001) but decreased the number of TUNEL^+^ NeuN^+^ cells (*F*_(2,9)_ = 446.739, *p* < 0.001) in the injured cortex (Figure 2A–C). Similarly, agomir-331-treated mice exhibited increased NeuN^+^ neurons (*F*_(2,9_) = 100.42, *p* < 0.001) and decreased TUNEL^+^ NeuN^+^ cells (*F*_(2,9_) = 147.015, *p* < 0.001) in the injured hippocampus at 3 dpi (Figure 2D–F). These data suggested that agomir-331 plays a neuroprotective role in the acute phase of HSI.

### 3.3. Agomir-331 Inhibits Glial Scar Formation in HSI Mice

Glial scar formation has long been recognized as a major impediment to neuronal regeneration [2,3]. Next, we evaluated whether agomir-331 regulates the glial scar formation following HSI by immunostaining brain sections with anti-GFAP and anti-Iba1 antibodies. We observed that the densities of both GFAP^+^ astrocytes (*F*_(2,9)_ = 178.324, *p* < 0.001) and Iba1^+^ microglia/macrophage (*F*_(2,9)_ = 367.309, *p* < 0.001) were dramatically reduced around the injury site in the agomir-331-treated cortex at 7 dpi (Figure 3A–C). Similarly, the numbers of astrocytes (*F*_(2,9)_ = 1463.092, *p* < 0.001) and microglia/macrophage (*F*_(2,9)_ = 181.446, *p* < 0.001) were also significantly decreased around the injury site of the hippocampus with the agomir-331 treatment, as evidenced by smaller sizes of GFAP^+^ and Iba1^+^ areas in the agomir-331 group compared to that of the sham group (Figure 3D–F). These results suggested that agomir-331 inhibits glial scar formation in the acute phase of HSI.

To further determine whether agomir-331 acts on microglia or infiltrating macrophages, we performed immunohistochemical staining of Iba1 and CD11a, a panleukocyte marker that is expressed by lymphocytes, monocytes, macrophages, neutrophils, basophils, and eosinophils. We found that intranasal delivery of agomir-331 significantly reduced the numbers of both filtrating macrophage (CD11a^+^ Iba1^+^) and residual microglia (CD11a^−^ Iba1^+^) in the injured cortex (Figure 3G–I), as well as in the injured hippocampus at 7 dpi (Figure 3J–L).

### 3.4. Administration of Agomir-331 Suppresses Astrocyte Proliferation and Migration

Glial scarring is characterized by high levels of astrocytic proliferation and migration [31]. To further explore the mechanism of agomir-331 in glial scar formation, we examined the proliferation and migration of astrocytes with or without agomir-331 treatment in vitro. To mimic an HSI-like inflammatory environment, we treated primary microglia with LPS (100 ng/mL) to collect microglial conditioned medium (MCM). As expected, there was a huge reduction of miR-331 in both astrocytes (*F*_(1,6)_ = 3921.323, *p* < 0.001) and microglia (*F*_(1,6)_ = 24.885, *p* < 0.01) that were exposed to MCM for 24 h (Figure 4A), indicating that miR-331 was also down-regulated in astrocytes and microglia under pro-inflammatory conditions in vitro. Next, we transduced astrocytes with agomir-331 or scramble, then treated them with MCM or control medium for 24 h, followed by BrdU labeling for 6 h. As shown in Figure 4B, compared with the control medium group, MCM treatment did enhance the percentage of BrdU^+^GFAP^+^ cells among GFAP^+^ cells (*F*_(2,9)_ = 42.553, *p* < 0.001). In contrast to the scramble treatment, administration of agomir-331 significantly inhibited the proliferation of astrocytes under MCM conditions. Consistently, CCK-8 assay confirmed the reduced proliferative ability of agomir-331-treated astrocytes under MCM conditions (*F*_(2,9)_ = 126.233, *p* < 0.001) (Figure 4C).

To determine the effect of miR-331 on astrocyte migration, primary astrocytes were transduced with agomir-331 or scramble and subjected to scratch and MCM for a given period of time. Our results evidenced that MCM significantly increased the migration of primary astrocytes, and agomir-331 treatment inhibited the ability of astrocytes to invade the acellular area for 12 h (*F*_(2,9)_ = 7.829, *p* < 0.05), 24 h (*F*_(2,9)_ = 7.271, *p* < 0.05) and 48 h (*F*_(2,9)_ = 5.911, *p* < 0.05), relative to cells treated with scramble and MCM (Figure 4D). Together, these results suggested that agomir-331 suppresses the proliferation and migration of astrocytes under cell stress conditions.

### 3.5. Agomir-331 Suppresses Inflammatory Response in MCM-Treated Microglia

Microglia-driven inflammation is pivotal in chronic neurodegeneration following TBI [32,33]. We therefore explored whether agomir-331 could modulate the inflammatory responses in microglia. Remarkably, agomir-331 enhanced the release of anti-inflammatory molecules *Arg-1* (*F*_(2,9)_ = 118.623, *p* < 0.001) and *IL-10* (*F*_(2,9)_ = 212.821, *p* < 0.001), but suppressed the expression of pro-inflammatory factors *TNF-α* (*F*_(2,9)_ = 8.751, *p* < 0.01), *IL-1β* (*F*_(2,9)_ = 498.564, *p* < 0.001), *iNOS* (*F*_(2,9)_ = 87.343, *p* < 0.001), and *IL-6* (*F*_(2,9)_ = 241.985, *p* < 0.001) in MCM-treated microglia (Figure 5A,B). These findings indicated that agomir-331 can effectively reduce inflammation, again supporting the finding that agomir-331 plays a neuroprotective role after brain injury.

### 3.6. Agomir-331 Improves Learning and Memory in HSI Mice

Since agomir-331 reduced the glial scar formation and protected against neuronal apoptosis following HSI, we then performed the Barnes maze and Y-maze tests to evaluate the learning and cognitive ability of HSI mice with or without agomir-331 treatment. In the Barnes maze test, agomir-331-treated HSI mice displayed significantly reduced latencies to locate the target hole in both training *(F*_(2,21)_ = 6.365, *p* < 0.01) and probe phases *(F*_(2,21)_ = 20.098, *p* < 0.001), compared with scramble-treated HSI mice (Figure 6A–C). Consistently, agomir-331-treated mice spent a longer time in the target quadrant compared to scramble-treated mice after HSI in the probe trial of the Barnes maze test (*F*_(2,21)_ = 14.32, *p* < 0.001) (Figure 6D). However, agomir-331 treatment did not impact the mean walking speed in the probe test compared to scramble treatment (Figure 6E).

In the subsequent Y-maze test, although there were no differences in the number of arm entries and in the mean speed between groups (Figure 6F–H), agomir-331-treated mice exhibited an increased spontaneous alternation performance compared with that of scramble controls (*F*_(2,21)_ = 5.586, *p* < 0.05) (Figure 6I). Altogether, these behavioral results provide robust evidence that agomir-331 improves cognitive function in HSI mice.

### 3.7. IL-1β Is a Direct Target Gene of miR-331

To investigate the downstream targets of miR-331, the TargetScan database was utilized to predict the potential targets of miR-331. Of interest, the key proinflammatory cytokine IL-1β has a predicted binding site of miR-331 on its 3′-UTR (Figure 7A).

To determine whether IL-1β is a direct target of miR-331, we cloned the wild-type or mutant 3′-UTR of IL-1β into a dual luciferase reporter construct, which allowed us to examine IL-1β translation by measuring luciferase activities. We found that agomir-331 could significantly inhibit the expression of firefly luciferase (*F_(_*_3,12)_ = 6.265, *p* < 0.01)through the wild-type but not mutant 3′-UTR of IL-1β (Figure 7B). ELISA analysis further revealed that the protein expression of IL-1β was dramatically decreased in the injured hippocampus with the treatment of agomir-331 (*F*_(2,9)_ = 11.24, *p* < 0.01), compared with that in the scramble-treated hippocampus at three days after HSI (Figure 7C). These results supported the finding that miR-331 directly binds to and silences IL-1β.

## 4. Discussion

TBI triggers glial scar formation and an inflammatory cascade, which usually lead to neurodegeneration and neurological dysfunction [34]. Manipulating these pathological processes is essential for brain repair. However, the modulators that regulate glial scaring and neuroinflammation are largely unknown. Glial scar is a highly complex system of interacting astrocytes, microglia, and other cell types. Determining how to manipulate this complex system is still an unresolved challenge. This study demonstrated that miR-331 expression was decreased early after TBI. Intranasal delivery of agomir-331 reduced glial scarring, neuroinflammation, and neuronal loss while improving cognitive function in a mouse TBI model. These findings highlight the potential therapeutic benefits of administering synthetic miR-331 mimics for TBI treatment.

Reactive astrocytes and microglia are hallmarks of TBI and the main cellular component of glial scar. Microglia are immediately activated upon brain injury and become main executors in neuroinflammation. In the early stages of TBI, microglia secrete pro-inflammatory molecules (IL-1β, TNF-α, IL-6, and iNOS) which are considered a vital driver of brain damage [35]. Meanwhile, the activated astrocytes actively propagate and up-regulate the expression of soluble factors (IL-1a, C1q, and TNF-α) which amplify the inflammatory process and exacerbate brain damage [36,37,38]. IL-1β is a crucial regulator of inflammation with increased levels after TBI [39,40]. IL-1β induces astrocyte proliferation and promotes astrocyte migration [41,42]. It is also noteworthy that IL-1β is a pivotal contributor to the neurodegenerative process [43], and inactivation of IL-1β relieves inflammation and neurotoxicity in rodents [44,45]. Current clinical strategies targeting IL-1β include canakinumab (a therapeutic monoclonal antibody targeting IL-1β) and anakinra (an IL-1 receptor antagonist, IL-1Ra) [46,47]. Additionally, several preclinical studies suggest that the blockade of IL-1β through anti-IL-1β or neutralizing antibodies attenuate inflammatory responses, brain tissue loss, and cognitive deficits following TBI in mice [48,49,50]. However, high manufacturing costs often hinder the application of protein-based drugs, leading to the pursuit of alternative and more cost-effective strategies [51]. Our study identifies IL-1β as a direct downstream target of miR-331, and nasal delivery of agomir-331 can efficiently inhibit the upregulation of IL-1β following TBI. Thus, these findings indicate that miR-331 mimics provide a promising alternative for IL-1β-targeted therapies.

Of note, this preliminary study has several limitations. Firstly, all presented data in the present study were collected in a short time window after TBI, and it is well known that TBI is an evolving condition in which secondary degeneration progresses over weeks in both animal models and clinical research. Certainly, there is a need to confirm the therapeutic potential of miR-331 mimics in long-term studies in vivo. Secondly, to avoid possible effects of the phase of female estrous cycle on behaviors [52], we only used male mice in the present study. Since there is a growing body of clinical and laboratory evidence proving the neuroprotective roles of estrogen and progesterone in TBI [53,54,55], it would be interesting to use age-matched male and female mice to examine whether there is a potential sex-dependent effect of mir-331 on neuroinflammation and glial scarring. Thirdly, a single miRNA potentially targets hundreds of genes that are involved in a functional interacting network [56,57]. Therefore, we cannot rule out the possibility that miR-331 protects against brain damage by regulating not only IL-1β but also other downstream targets. Future studies are needed to explore the regulatory network of miR-331 by combined transcriptional and translational profiling. Fourthly, given that intranasal administration cannot specifically deliver drugs to the brain [58], techniques (such as nanotechnology) that allow brain-cell-type-specific delivery of miR-331 mimics remain to be developed. Finally, although early intervention has been suggested to be critical for improving functional outcomes of TBI patients [59,60], it is simply not possible to administer medication to patients immediately after the injury. The incidence of penetrating traumatic brain injury (TBI) is relatively low among all TBI types, and many miRNAs display dynamic and specific temporal and spatial expression patterns following TBI [21,61]. Therefore, future investigations considering cell-type-specific delivery and systemic administration of multiple miRNA agomirs and/or antagomirs in both focal and diffuse TBI animal models are required before translating research findings into clinical practice.

In conclusion, our study suggests that agomir-331 is a repressor of reactive gliosis and neuroinflammation by inhibiting the expression of IL-1β following TBI. If miR-331 demonstrates efficacy in humans, the storage and delivery methods for this therapeutic should be optimized to enable prompt administration to patients with brain injuries in real-world settings. Specifically, the formulation and stability of miR-331 must facilitate rapid treatment following traumatic brain injury to ensure maximal therapeutic benefit. Considering that IL-1β perpetuates immune responses and contributes to disease severity in stroke, multiple sclerosis, neurodegenerative diseases, and diabetic retinopathy [62], it would be interesting to examine whether application of miR-331 mimics is also beneficial for enhancing neuronal recovery in these CNS diseases.

## Figures and Tables

**Figure 1 cells-12-02429-f001:**
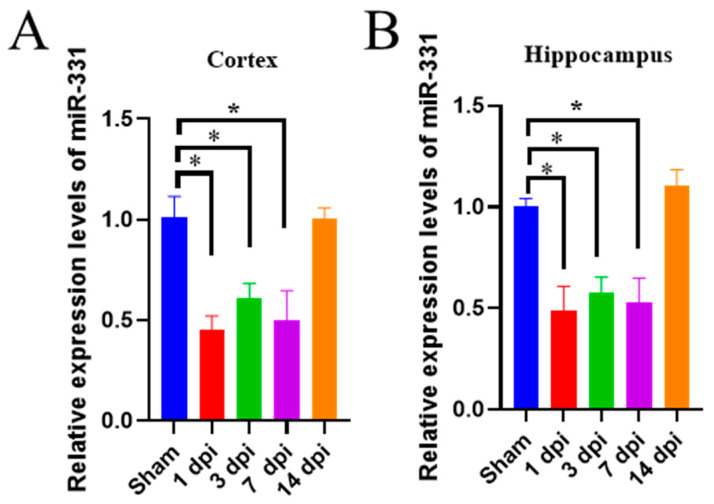
The expression of miR-331 is decreased in the early stages following HSI. (**A**) Relative mRNA expression levels for miR-331 in cortex after HIS. (**B**) Relative mRNA expression levels for miR-331 in hippocampus after HSI. *n* = 4; ** p* < 0.05.

**Figure 2 cells-12-02429-f002:**
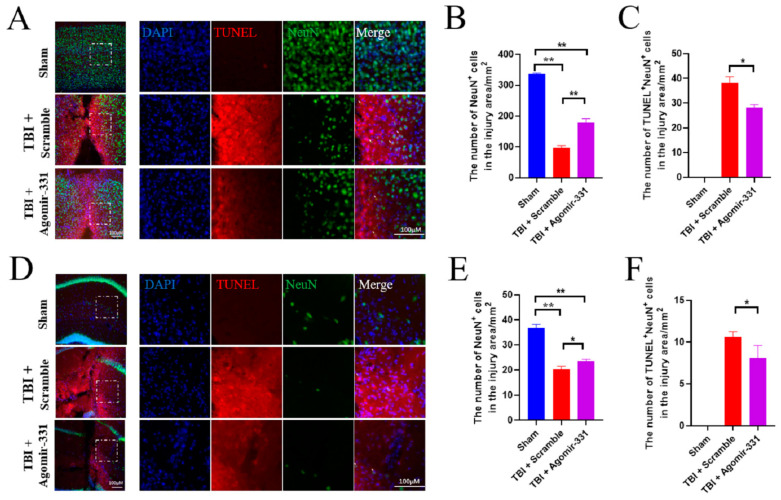
Agomir-331 inhibits neuronal apoptosis after HSI. (**A**) Representative images showing neuronal apoptosis in the cortex at 3 dpi by immunofluorescence staining for NeuN (green) and TUNEL (red). The regions within the dotted white boxes (**left panel**) are shown in a higher magnification view on the right panel. Scale bars, 100 μm. (**B**,**C**) Quantifications of the number of NeuN^+^ neurons (**B**) and the percentage of NeuN^+^ TUNEL^+^ cells (**C**) in the injured area of the cortex at 3 dpi. (**D**) Representative images showing NeuN^+^ and TUNEL^+^ cells in the hippocampus at 3 dpi. The regions within the dotted white boxes (**left panel**) are shown in a higher magnification view on the right panel. Scale bars, 100 μm. (**E**,**F**) Quantifications of neurons (**E**) and NeuN^+^ TUNEL^+^ cells (**F**) in the hippocampus at 3 dpi. *n* = 4; ** p* < 0.05, *** p* < 0.01.

**Figure 3 cells-12-02429-f003:**
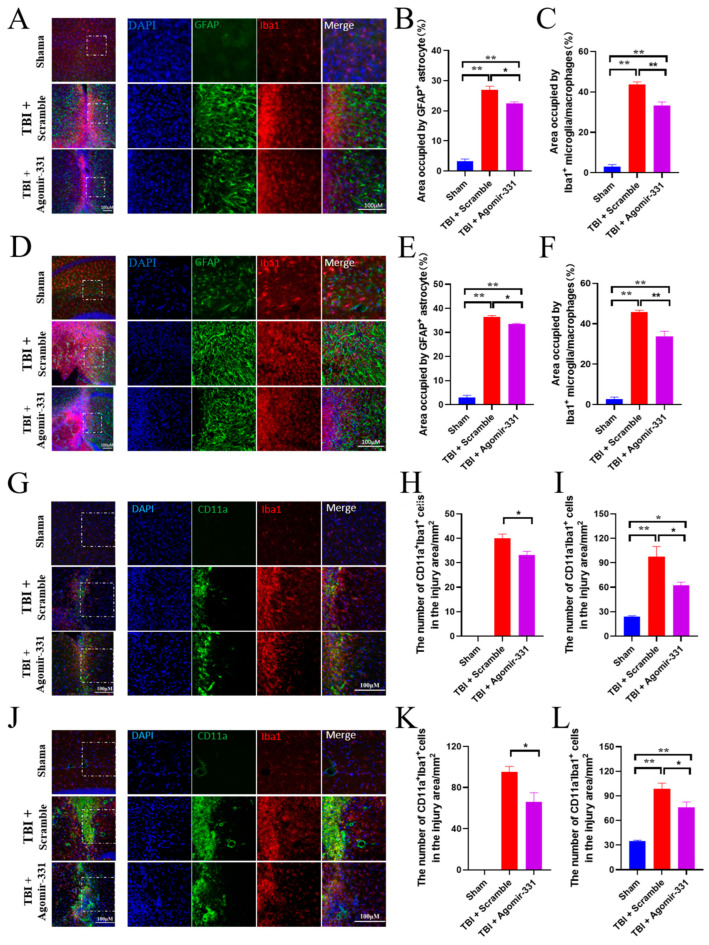
Agomir-331 suppresses HSI-induced astrogliosis and microglial activation. (**A**) Representative images of immunohistochemical staining of GFAP (green) and Iba1 (red) in the cortex at 7 dpi. The regions within the dotted white boxes (**left panel**) are shown in a higher magnification view on the right panel. Scale bars, 100 μm. (**B**,**C**) Quantification of the areas of GFAP (**B**) and Iba1 (**C**) around the injury site in the cortex at 7 dpi. (**D**) Representative images showing the activation of microglia/macrophage in the hippocampus at 7 dpi. The regions within the dotted white boxes (**left panel**) are shown in a higher magnification view on the right panel. Scale bars, 100 μm. (**E**,**F**) Quantification of the areas of GFAP (**E**) and Iba1 (**F**) around the injury site in the hippocampus at 7 dpi. (**G**) Representative images of immunohistochemical staining of CD11a (green) and Iba1 (red) in the cortex at 7 dpi. The regions within the dotted white boxes (**left panel**) are shown in a higher magnification view on the right panel. Scale bars, 100 μm. (**H**,**I**) Quantification of the numbers of CD11a^+^ Iba1^+^ cells (**H**) and CD11a^−^ Iba1^+^ cells (**I**) around the injury site in the cortex at 7 dpi. (**J**) Representative images showing the activation of microglia/macrophage in the hippocampus at 7 dpi. The regions within the dotted white boxes (**left panel**) are shown in a higher magnification view on the right panel. Scale bars, 100 μm. (**K**,**L**) Quantification of the areas of CD11a^+^ Iba1^+^ (**K**) and CD11a^−^ Iba1^+^ cells (**L**) around the injury site in the hippocampus at 7 dpi. *n* = 4; ** p* < 0.05, *** p* < 0.01.

**Figure 4 cells-12-02429-f004:**
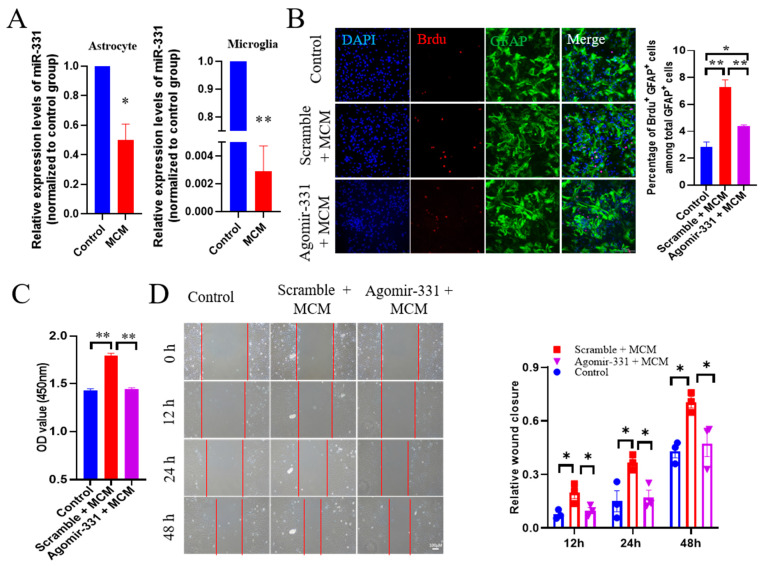
Agomir-331 inhibits astrocyte proliferation and migration under MCM conditions in vitro. (**A**) Relative expression levels for miR-331 in primary astrocytes and microglia treated with MCM for 24 h. (**B**) Representative images and quantification of GFAP^+^ and BrdU^+^ cells subjected to agomir-331 treatment and MCM conditions. Scale bar, 200 μm. (**C**) Optical density (OD) values of agomir-331- and MCM-treated astrocytes determined by CCK-8 assay. (**D**) Representative images and quantification of the migration of astrocytes treated with agomir-331 and MCM in the scratch assay, an in vitro model of TBI. Scale bars, 100 μm. *n* = 3 - 4; ** p* < 0.05, *** p* < 0.01.

**Figure 5 cells-12-02429-f005:**
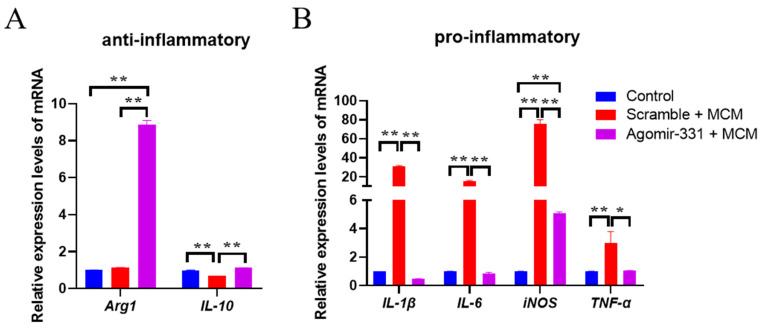
Agomir-331 inhibits inflammatory response in MCM-stimulated microglia. (**A**) Relative mRNA expression levels of *Arg1* and *IL-10* in primary microglia treated with agomir-331 and MCM by qPCR. (**B**) The mRNA levels of *IL-1β*, *IL-6*, *iNOS*, and *TNF-α* in primary microglia treated with agomir-331 and MCM. Data are represented as means ± SEM. *n* = 4; ** p* < 0.05, *** p* < 0.01.

**Figure 6 cells-12-02429-f006:**
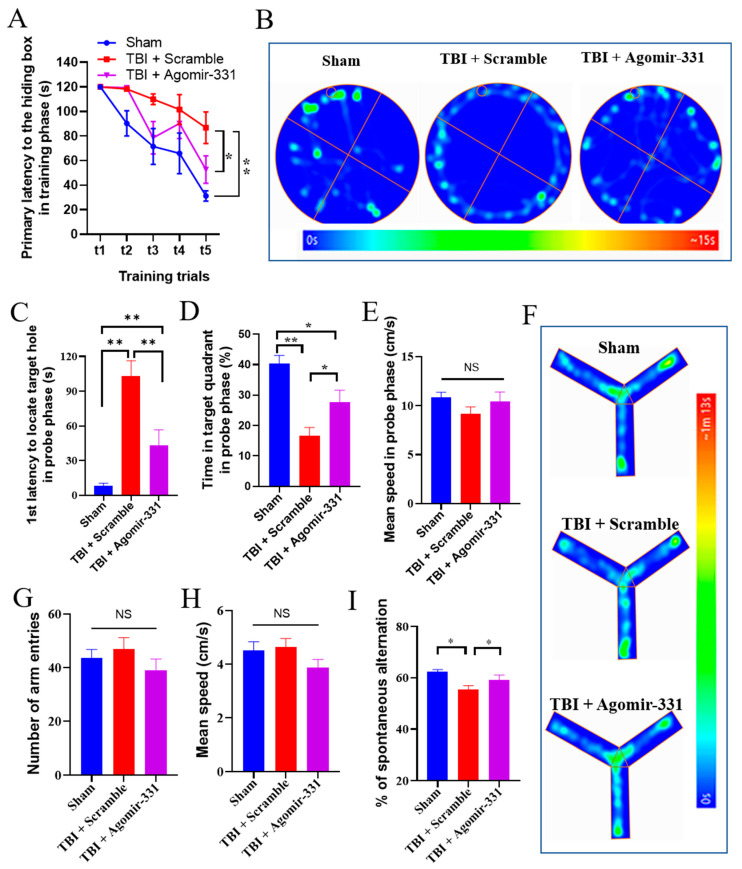
Agomir-331 attenuates the cognitive deficits in HSI mice. (**A**) Escape latencies for each group of mice to reach the hiding box in training phase of the Barnes maze test (BMT). (**B**) Representative heatmaps for the BMT probe test. The red circle indicates the location of the escape box in the target hole. (**C**) First latency to locate target hole in probe phase of BMT. (**D**) Time spent in target quadrant in probe phase of BMT. (**E**) Mean moving speed in probe phase of BMT. (**F**) Representative heatmaps for the Y-maze test. (**G**) Total numbers of arm entries during Y-maze exploration. (**H**) Mean moving speed in the Y-maze test. (**I**) Percentage of spontaneous arm alternation during Y-maze testing. *n* = 7–8 mice per group; ** p* < 0.05, *** p* < 0.01. NS, non-significant.

**Figure 7 cells-12-02429-f007:**
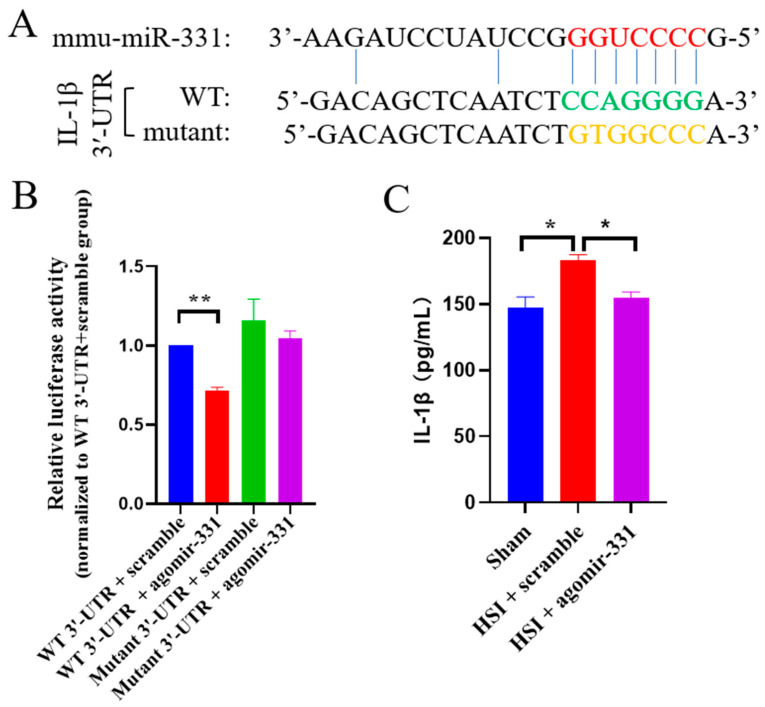
IL-1β is a direct downstream target of miR-331. (**A**) Alignment of miR-331 germline region with IL-1β 3′UTR and complementary site mutation of IL-1β 3′UTR sequence. (**B**) HEK293 cells were co-transfected with a luciferase reporter vector expressing WT or mutant IL-1β 3′UTR, and luciferase activity was measured. (**C**) The protein expression of IL-1β in the agomir-331-treated brain was analyzed at 72 h after HSI via ELISA. Values are means ± SEM. Significance was determined by one-way ANOVA and Tukey’s post hoc analysis, *n* = 3 - 4; * *p* < 0.05; ** *p* < 0.01.

**Table 1 cells-12-02429-t001:** Primers used for qRT-PCR.

Gene		Primer Sequence (5’-3’)
*Actin*	*Forward*	*TGCACCACCAACTGCTTAG*
	*Reverse*	*GGATGCAGGGATGATGTTC*
*TNF-α*	*Forward*	*ACGGCATGGATCTCAAAGAC*
	*Reverse*	*GTGGGTGAGGAGCACGTAGT*
*IL-1β*	*Forward*	*CAGGCAGGCAGTATCACTCA*
	*Reverse*	*TGTCCTCATCCTGGAAGGTC*
*iNOS*	*Forward*	*GGCAGCCTGTGAGACCTTTG*
	*Reverse*	*GCATTGGAAGTGAAGCGTTTC*
*Arg1*	*Forward*	*ACATTGGCTTGCGAGACGTA*
	*Reverse*	*TCCATCACCTTGCCAATCCC*
*IL-6*	*Forward*	*ATGGATGCTACCAAACTGGAT*
	*Reverse*	*TGAAGGACTCTGGCTTTGTCT*
*IL-10*	*Forward*	*CCCTGGCTCGTGTGGATTT*
	*Reverse*	*GACCGATACCACTCCTCTGTC*

## Data Availability

All datasets supporting the conclusions are included in the article. Further enquiries on data and materials can be directed to the corresponding author.

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
