# Peer review of "Agomir-331 Suppresses Reactive Gliosis and Neuroinflammation after Traumatic Brain Injury"

_cells, 2023, doi:10.3390/cells12202429_

Round 1
Reviewer 1 Report
This study proposes that mir-331 is a pivotal regulator of reactive gliosis and neuroinflammation after a traumatic brain injury. The word 'pivotal' is not warranted, the effects presented are of very moderate impact overall. Furthermore, the text needs to be improved at multiple levels:
a) there is confusion in the use of terms - the authors talk about glial scarring, when the only thing they show is a quantification of the GFAP staining. The glial scar is a much more complex entity. The authors use Iba1 staining to analyse microglia, but they do not show awareness that Iba1 also stains infiltrating macrophages.
b) in the methods, there is a need for a clearer description of the size of the lesion: the AP and ML coordinates appear to indicate a very large lesion for a mouse brain, and there is also concern that this is a type of lesion which poorly represents human traumatic brain injury. The dose of agomir-331 should be expressed in total amount administered over the repeated applications. The locations of the brain sections analysed by immunofluorescence , vs the site of the lesion, is not clearly indicated. In the wound healing assay, how was the wound close quantified? It is also not clear when the behavioural tests were carried out, after injury. And how was the hippocampal and cortex tissue dissected for the qt-PCR analysis?
c) the images of immunostaining are of poor quality and it is not clear where they came from in terms of location of the image; there is a need for higher magnification images (and clear size bars) and indication of the location chosen for the representative images shown.
d) the quotation of references is incorrect in certain instances, e.g. line 289 on page 9 and line 361, page 11. More care is required, so that references truly support the point made.
e) a general point is how relevant the administration regime is: the first administration of agomir-331 occurs immediately after injury, and this is simply not possible in the context of real life human injuries. If the very first immediate application is omitted, the effect may not be seen at all. So this raises some serious doubts as to the potential therapeutic value of this approach. The authors should qualify their concluding statements accordingly.
Fine, overall, with some checks for typos/spelling required.
Reviewer 2 Report
The authors have investigated the role of mir-331 in neuroinflammation and glial scarring. This interesting study can be made much better by significantly increasing scientific rigor and reproducibility. The authors must use age matched male and female mice to investigate the role of mir-331 in neuroinflammation and glial scarring. Like most microRNAs, mir-331 also targets several genes including Sequestosome 1 and Optineurin. Authors must test multiple genes which play a crucial role in neuroinflammtion , neurodegeneration and glial scarring. Authors must perform additional motor and behavioral studies including Rotarod, novel object recognition, elevated plus maze, fear conditioning as well as beam balance and beam walk well spaced out throughout the duration of the study. Provide NSS scores and all behavioral studies must be accompanied by heatmaps. Immunofluorescence microscopy appears to be biased. It must show the ipsilateral and contralateral sides including the hippocampal and CA regions in sham, TBI, TBI+Scramble and TBI+Agomir-331. In addition to NeuN, IBA1 and GFAP authors must perform immunostaining for the markers of neuroinflammation, NLRP3, complement, neurodegeneration (TDP43, Beta amyloid, Tau) as well neurogenesis (SOX2, DCX, Nestin). Glial scarring is a late stage event and authors must analyze male and female mice brains at not only early but also at multiple later time points using immunofluorescence microscopy. The quality of Figure 3 is not at all impressive.
There is definitely a significant scope to improve upon different sections of the manuscript. Please provide specific statements regarding the current knowledge gap, hypothesis, rationale and potential benefits.
Reviewer 3 Report
In this study, the possible role of miR-331 is analyzed in mouse model of hippocampal stab injury (HIS). In particular, the intranasal administration of agomir-331, an upgraded product of miR-331 mimics, suppressed reactive gliosis and neuronal apoptosis and improved cognitive function in HSI mice. Authors also identified IL-1β as a direct downstream target of miR-331 starting from a bioinformatic approach (TargetScan database), then using a dual luciferase reporter construct to examine IL-1β protein translation in HEK293 cells
The main weakness of the study is the lack of a statistical design, regarding both in vivo (no power analysis for sample size calculation is presented) and in vitro experiments (how many independent experiments? How many replicates in each experiment?).
At the end, n=7-8 mice were included in the behavioral studies, n=3-4 mice in RT-PCT and n=? mice in the immunofluorescence staining.
Other key information are missing as:
- At which post-TBI time were carried out behavioral tests?
- Number of sections analyzed/mice and ROI dimension for results include in Figure 3
- How dosage and agomir-331 diffusion after intranasal administration are monitored?
As also discussed by the authors, MiRNAs may target several different gene transcripts and their regulation is often redundant among different miRNAs. Thus, the final meaning of the study is weak
In conclusion: this is a collection of promising results (pilot study), which must be substantiated through an appropriate statistical design of each experiment
appropriate
Round 2
Reviewer 2 Report
While the authors have made an attempt to revise the manuscript the major concerns still remain. Scientific rigor and reproducibility are still a major concern, The authors have failed to include female mice and have not performed the crucial experiments suggested earlier. Immunofluorescence images are selectively chosen and do not represent the same region across the control, TBI+Scarmble as well as TBI+Agomir-331. Authors must include TBI+Antagomir-331 to conclusively confirm and validate their current research findings. Additional behavioral experiments, immunofluoresecence analysis, lesion volume, cerebral edema, blood brain barrier dysfunction must be thoroughly investigated and data must be provided.
Extensive editing of English language is absolutely necessary. Even the current title is wrong. There are several mistakes. For example: administered through nasal by pipette, hippocampal tissue were cut, behavioral data 378 strongly support that agomir-331, GFAP and Iba1 expressions were etc. There are very large sentences such as: Considering that the incidence of penetrating injury is relatively low among all types of TBI, and many miRNAs display dynamic and specific temporal and spatial patterns of expression following TBI [21, 57], future investigations considering different sexes, cell type-specific delivery and systemic administration of a cocktail of multiple agomirs and/or antagomirs in both focal and diffuse TBI animal models are required before translating research findings into clinical practice.
Reviewer 3 Report
I thank the Authors for the effort in providing all required statistic details.
I would recommend including a statement, as study limitation, regarding the need to confirm the hypothesis suggested by these preliminary experiments in long term studies in vivo. In fact, all presented data are collected in a short time-window after TBI, and is well known that TBI is an evolving condition, in which secondary degeneration progress over weeks also in animal models.
Adeguate, minor typos
Author Response
Comments and Suggestions for Authors: I thank the Authors for the effort in providing all required statistic details. I would recommend including a statement, as study limitation, regarding the need to confirm the hypothesis suggested by these preliminary experiments in long term studies in vivo. In fact, all presented data are collected in a short time-window after TBI, and is well known that TBI is an evolving condition, in which secondary degeneration progress over weeks also in animal models.
Response: Many thanks for the positive comments as well as these invaluable recommendations. We have included a statement that described the time-window limitation of the present study in the revised manuscript (Lines 452-486).
Comments on the Quality of English Language: Adeguate, minor typos.
Response: We have proofread the manuscript and corrected typos with a friend who is a native English speaker.